# Killer Cell Immunoglobulin-like Receptor Genotypes and Reproductive Outcomes in a Group of Infertile Women: A Romanian Study

**DOI:** 10.3390/diagnostics13193048

**Published:** 2023-09-25

**Authors:** Mihai Surcel, Iulia Adina Neamtiu, Daniel Muresan, Iulian Goidescu, Adelina Staicu, Monica Mihaela Marta, Georgiana Nemeti, Radu Harsa, Bogdan Doroftei, Mihai Emil Capilna, Gabriela Caracostea

**Affiliations:** 11st Department of Obstetrics and Gynecology, “Iuliu Hatieganu” University of Medicine and Pharmacy, 3-5 Clinicilor Street, 400347 Cluj-Napoca, Romania; surcel.mihai75@gmail.com (M.S.); daniel.muresan@umfcluj.ro (D.M.); iuliangoidescu@gmail.com (I.G.); adelina.staicu@umfcluj.ro (A.S.); georgiana_nemeti@yahoo.com (G.N.); caracostea1@yahoo.com (G.C.); 2Health Department, Environmental Health Center, Part of ALS, 58 Busuiocului Street, 400240 Cluj-Napoca, Romania; 3Faculty of Environmental Science and Engineering, Babes-Bolyai University, 30 Fantanele Street, 400294 Cluj-Napoca, Romania; 4Department of Medical Education, “Iuliu Hatieganu” University of Medicine and Pharmacy, 3-5 Clinicilor Street, 400347 Cluj-Napoca, Romania; mmarta@umfcluj.ro; 5In Vitro fertilization Department, “Regina Maria” Hospital, 29 Dorobantilor Street, 400117 Cluj-Napoca, Romania; raduharsa@gmail.com; 6Faculty of Medicine, “Grigore T. Popa” University of Medicine and Pharmacy, 16 University Street, 700115 Iasi, Romania; bogdandoroftei@gmail.com; 71st Department of Obstetrics and Gynecology, University of Medicine, Pharmacy Science and Technology “George Emil Palade”, 38 Gheorghe Marinescu, 540142 Targu Mures, Romania; mcapilna@gmail.com

**Keywords:** endometrium disorder, killer cell immunoglobulin-like receptor polymorphism, infertility, NK cells, Romania

## Abstract

A growing body of evidence suggests that endometrial immune disorders may be responsible for endometrial dysfunctions that can lead to gynecological and obstetrical pathology. The aim of this study was to explore the potential relationship between different killer cell immunoglobulin-like receptor (KIR) genotypes and reproductive outcomes. We conducted a prospective cohort study that included 104 infertile patients undergoing an in vitro fertilization procedure. All participants underwent clinical and ultrasound examination, genetic evaluation (KIR genotyping), endometrial washing fluid sampling for cytokine determination, endometrial tissue sampling for histologic assessment and hysteroscopic evaluation. Our analysis showed statistically significant lower levels of uterine cytokines TNF-α (*p* = 0.001) and IL-1beta (*p* = 0.000) in the KIR AA genotype group as compared to KIR AB and BB among study participants with chronic endometritis. The study results suggest that the KIR AA genotype population subgroups may be more susceptible to developing endometrial disorders such as chronic endometritis. The changes in the behavior of NK cells seem to be subtle and expressed as an altered regulatory pattern.

## 1. Introduction

Infertility is known as a vast and polymorphic medical condition which engages a massive medical effort [1], often with disappointing results. Among the disorders which are decreasing the fertility potential, endometrial pathology has a special role. Endometrial dysfunction is currently perceived as a preclinical condition preceding many of the gynecologic pathologies (e.g., recurrent implantation failure (RIF), endometriosis, adenomyosis, fibroma, polyps, endometrial neoplasm) [2], as well as an important contingent of obstetrical pathology such as abortion, preeclampsia, or intrauterine growth restriction [3]. The alterations in the endometrial maturation control mechanisms are a major contributor to the development of these pathologies [4].

Extensive research conducted in the past years provides evidence for the implication of the immune system in this process [5]. The NK cell plays a major role in the defense system due to its anti-tumor and antiviral activity. Besides its cytotoxic capacities, the uterine NK cell has varied and equally specific regulatory functions, being involved at the same time, in the tissue growth and also in modulating other immune cells [6].

In the endometrial mucosa, the uterine NK cell adapts to the needs of this tissue through intercellular communication, and performs very specific actions such as apoptosis of distinct cellular populations or uterine vascular remodeling [7,8]. The uterus is probably a suitable territory for the NK cells, where they go through a rapid development, evolving from agranulated cells in the early proliferative phase, to very active cells in the late luteal phase and especially in the first weeks of gestation [9]. If the NK cell is currently assumed to be a killer, at this particular level (the uterus), it converts into a major regulatory agent that retains less cytotoxic abilities [9,10].

The regulatory functions of the NK cells are exerted through a diverse production of cytokines, among which the following predominate: the production of granulocyte colony-stimulating factor (G-CSF), granulocyte-macrophage colony-stimulating factor (GM-CSF), macrophage colony stimulating factor (M-CSF), tumor necrosis factor-α (TNF-α), interleukins (IL) (e.g., IL-6, IL-8, IL-1B), leukemia inhibitory factor (LIF) or angiogenic factors such as vascular endothelial growth factor (VEGF), angiopoietin-1 (ANG-1), and angiopoietin-2 (ANG-2) [11,12]. All those factors contribute to the endometrium development and the recognition/tolerance of the embryo, as well as to the initiation of placentogenesis by coordinating the trophoblastic invasion [13].

The NK cell activity is determined by the receptors they express. Two major categories of receptors have been described on these cells: killer immunoglobulin-like receptors (KIR) that identify molecules belonging to the human leukocyte antigen (HLA) system (HLA-A; HLA-B; HLA-C) and CD94/NKG2A, a heterodimer that recognizes HLA-E [14]. KIR are type-1 transmembrane glycoproteins belonging to the immunoglobulin superfamily, and they are mainly expressed on NK cells, but to a small extent also on certain subtypes of T lymphocytes [15]. The KIR system includes fifteen genes and two pseudogenes described as KIR 2D or KIR 3D, depending on the number of extracellular domains (two or three, respectively): KIR2DL1, KIR2DL2, KIR2DL3, KIR2DL4, KIR2DL5A, KIR2DL5B, KIR2DS1, KIR2DS2, KIR2DS3, KIR2DS4, KIR2DS5, KIR3DL1, KIR3DL2, KIR3DL3, KIR3DS1, and as KIR-S or KIR-L, depending on the length of encoded cytoplasmic tail (short -S or long -L) [13]. Activating receptors present a short cytoplasmic fragment (S), while inhibitory receptors present a long cytoplasmic fragment (L). The KIR system is diverse, primarily due to allele polymorphism (over 5200 alleles are described), but also due to gene copy number variation (CNV). In this regard, the effort to limit the variations of haploid structures to certain preset patterns while still preserving the polymorphic concept has been essential for quantifying the NK cell activity [16].

The KIR genotypes have two haplotypes, A and B, defined based on their component genes. Haplotype A contains a fix number of genes: one activating receptor 2DS4 and eight inhibitory receptors KIR3DL3, 2L3, 2DP1, 2DL1, 3DP1, 2DL4, 3DL1, and 3DL2. Haplotypes B contains all the other genes (therefore having a variable set of genes) and at least one of the following activating genes: KIR2DS1, KIR2DS2, KIR2DS3, KIR2DS5 and KIR3DS1. Every individual possesses two haplotypes, resulting in three subgroups: KIR AA, KIR AB, and KIR BB. As patients with the heterozygote KIR AB exhibited an activatory profile, we assumed that the subgroups AB and BB were very similar, and both subgroups were included in one group (entitled as group B), while the group dominated by the inhibitory gene KIR AA was designated as group A.

All studies evaluating the A and B haplotypes in different cohorts confirmed the usefulness of this approach, but also documented multiple deviations from these patterns, following deletions/insertions or genetic recombination [15,17,18]. The first studies addressing the behavior of certain KIR genotypes targeted the obstetrics domain and identified a tendency towards more frequent obstetric complications (including placental insufficiency or preeclampsia) in the KIR AA variant group [19]. A series of mechanisms by which these pathologies are explained have been described, the main role being attributed to the angiogenesis process. Also, certain polymorphisms of some genes included in the KIR may be responsible for the development of other disorders such as endometritis and RIF [20,21].

Given the difficulty of quantifying the NK cell function and the reduced pathological relevance of their abundance in tissues, the evaluation of certain factors which define the functionality of the endometrium may be used in clinical practice as an indirect assessment method. Thus, altered endometrial receptivity, uterine flora imbalance, mild hysteroscopic alterations (congestion and micropolyps) or the occurrence of chronic endometritis could indirectly signal subtle changes in the behavior of NK cells, which usually remain undetected by standard testing.

The aim of this study was to explore the potential relationship between different KIR genotypes and reproductive outcomes (including chronic endometritis, abortion rate, endometrial quality (described by symptoms associated with endometrial pathology and hysteroscopic modifications), endometrial receptivity (quantified through LIF cytokine level) and the regulatory abilities of the NK cells (quantified through IL-1β and TNF-α cytokine levels).

## 2. Materials and Methods

### 2.1. Study Design and Participant Recruitment

We conducted a prospective cohort study within the Assisted Reproduction Department of the 1st Obstetrics and Gynecology Clinic (Cluj-Napoca, Romania), between May 2016 and June 2021. The study was conducted according to the guidelines of the Declaration of Helsinki and it was approved by the Ethics Committee of “Iuliu Hatieganu” University of Medicine and Pharmacy (protocol no. 352/2 June 2015)

The study included infertile women enrolled in an in vitro fertilization (IVF) procedure, who underwent hysteroscopy during the indicated time frame. The exclusion criteria included: (i) patients with endometrial disorders, such as endometriosis, adenomyosis, fibroma; and (ii) patients with intrauterine pathological structures detected on ultrasound examination. We approached 181 women, of which 17 refused to participate in the study. A detailed description of the study participants’ recruitment process is given in Figure 1. The rate of participation in the study was 90.6%. The participants were informed about the study aim and methodology. Informed consent was obtained from all women participating in the study, prior to their participation. Also, we collected medical history data from each participant at the time of enrollment in the study.

### 2.2. Biological Sample Collection and Analysis

The biological samples (blood and endometrial washing fluid) to be used for the determination of uterine cytokines and polymorphism testing were collected from the study participants before performing the hysteroscopy.

#### 2.2.1. Cytokine Analysis

A catheter was used to inject 2 mL of saline solution in the uterine cavity. The fluid was then aspirated and stored at −18 °C until analysis. Standard ELISA kit tests were used to determine IL-1β, TNF-α and LIF levels in the endometrial washing fluid. IL-1β and TNF-α levels were measured using IL-1β and TNF-α Human ELISA Kit (Thermo Fisher Scientific, Waltham, MA, USA), according to the manufacturer’s instructions. Also, LIF levels were measured using Human LIF SimpleStep ELISA Kit (Abcam, Cambridge, UK), according to the manufacturer’s instructions.

#### 2.2.2. Polymorphism Testing

Epicentre MasterPure Complete DNA and RNA Purification Kit (Illumina, Madison, WI, USA) was used for DNA extraction from blood. The blood samples were centrifuged for 15 min at 15,000× *g* to remove supernatant. The KIR genotype was determined using KIR-Ready Gene (Inno-train DIAGNOSTIK GMBH, Kronberg, Germany), according to the manufacturer’s instructions. The genotype identification was performed using the Allele Frequency Net Database [22].

The following genotype classification was used: genotype A, including genes 2DS4, KIR3DL3, 2L3, 2DP1, 2DL1, 3DP1, 2DL4, 3DL1, and 3DL2, and genotype B, including all group A genes and at least one additional KIR gene from the following genes: KIR2DS1, KIR2DS2, KIR2DS3, KIR2DS5 and KIR3DS1 (details are given in the Section 1).

### 2.3. Paraclinical Examinations

#### 2.3.1. Ultrasound Examination

All women enrolled in the study underwent transvaginal ultrasound examination to evaluate the uterus and annexes using a 5–7.5 MHz Medison Accuvix A30 Ultrasound System. Any focal lesions such as polyps, fibroids, intrauterine adhesions (IUA), retained products of conception, etc., were recorded. Endometrial thickness was measured in mid-sagittal plane and at the point of maximum thickness of the stripe.

#### 2.3.2. Hysteroscopy

Patients were scheduled for hysteroscopy on days 22–25 of their ovarian cycle. All hysteroscopies were performed by the same surgeon. Uterine distension was induced with saline solution, using the continuous-flow and pressure-controlled pump systems. During hysteroscopy, the inspection of the cervix, endocervical canal, uterine cavity, tubal ostia and endometrium was performed. The pathological findings were described according to the current nomenclature as: hypervascularization, mucosa elevation, micro-polyps, pale endometrium, endometrial defect, single adhesion band, uterine cavity abnormality, dysmorphic (arcuate) cavity, hemi-uterus, endometrial polyp(s), and fibroid(s). We counted the number of pathological findings per patient and assumed that an increased number of these modifications might be associated with an increased alteration of the endometrium.

A biopsy was also performed for the identification of the transmembrane heparin sulfate proteoglycan syndecan-1 (CD138) (a specific marker of plasma cells). The CD138-positive plasma cells were counted in 10 non-overlapping random stromal areas visualized at 400-fold magnification. The anti-CD138 monoclonal antibody clone EP 201 (Vitro Master Diagnostica, Seville, Spain) were used for the analysis. At least 20 high-power fields were examined per specimen. The test was considered positive for chronic endometritis if more than five CD138-positive cells were present, and negative if fewer than five positively stained plasma cells were present. All endometrial biopsy specimens were examined by the same histopathologist.

### 2.4. Statistical Analysis

The statistical analysis was performed using STATA v.18 statistical software (STATACorp LLC, College Station, TX, USA). We used mean, standard deviation, median and interquartile range to statistically describe the continuous variables and frequencies/percentages to describe the categorical variables. We tested the clinical and paraclinical factors for normal distribution using the Skewness and Kurtosis and Shapiro–Wilk tests, and considered those with *p* > 0.05 to be normally distributed. We used the Pearson chi2 and the Wilcoxon–Mann–Whitney to test for the differences between two genotype groups. We also tested for correlation between clinical (chronic endometritis) and paraclinical factors such as cytokine levels, hysteroscopic findings, and previous abortions, using the Pearson test. The statistical significance was defined as *p* < 0.05 for a two-tailed test.

## 3. Results

Table 1 shows the distribution of demographic, clinical, and paraclinical factors among our participants. The study included 104 female participants with ages between 29 and 43 years, most of them with idiopathic infertility diagnosis, from both genotype groups, and with a duration of infertility between two and ten years. Most of the study participants were nonsmokers and there was no statistically significant difference between the KIR AA and the KIR AB and BB group as regards the smoking status (*p* = 0.486). Most participants (in both genotype groups) did not have any clinical symptoms and there was no statistically significant difference between the KIR AA and the KIR AB and BB group participants with or without symptoms (*p* = 0.933). The hysteroscopic examination showed between zero and five abnormal findings in the KIR AA group, and between zero and three abnormal findings in the KIR AB and BB group. Figure 2 shows a normal hysteroscopic aspect of the endometrium and certain abnormalities identified during hysteroscopy in some of the study participants. Also, the KIR AA group included a significantly higher percent of patients with chronic endometritis (*p* = 0.002) and had no statistically significant (*p* = 0.122) lower levels (both mean and median concentration) of LIF in the endometrium washing fluid as compared to KIR AB and BB group.

The Wilcoxon–Mann–Whitney test showed statistically significant lower median concentration levels of uterine cytokines TNF-α (*p* = 0.001) and IL-1 β (*p* = 0.000) in the KIR AA genotype group as compared to KIR AB and BB, among study participants with chronic endometritis (Table 2).

The statistical analysis showed significant correlations between chronic endometritis and the levels of TNF-*α*, IL-1β, and LIF cytokines (*p* < 0.0001), and also between chronic endometritis and hysteroscopic findings (*p* < 0.0001). In addition, the modifications detected during the hysteroscopic examination were positively and significantly correlated with uterine levels of IL-1β cytokine (*p* = 0.010) and with previous abortions (*p* = 0.018) (Table 3).

## 4. Discussion

In our study, participants with the KIR AA genotype presented abnormalities suggesting a diminished fertility potential: a longer span of infertility and higher miscarriage rates, as well as a higher proportion of hysteroscopic abnormalities. On the other hand, in our study, clinical symptoms or endometrial thickness have not been accurate factors to be used for the identification of chronic endometritis or associated with pathological hysteroscopic modifications. Our results suggest that the KIR AA group may be more prone to developing certain gynecologic conditions. Knowing the multiple mechanisms through which the NK cell may be involved in the development of adenomyosis /endometriosis (e.g., cell recognition process, integrin production, and angiogenic abilities), we can speculate that the hysteroscopic modifications identified in the KIR AA patients may be regarded as stages that precede the development of these pathological conditions [16,20,21]. In this regard, the abnormal hysteroscopic images (mainly endometrial congestion, attributed to chronic endometritis), were frequently described in previous studies in patients with adenomyosis or endometriosis [1,2]. There is substantial evidence regarding the multifactorial and polygenic nature of many uterine disorders [23]. In the development of these disorders, it was implied that there are several functional endometrial mechanisms which are altered (e.g., cell cycle control, immune response, cellular metabolism and/or differentiation, angiogenesis, or the control of chromosomal stability, and, more recently, processes related to protein degradation or mitochondrial function), and they are exhibited as endometrial dysfunctions [24]. A recent large-scale meta-analysis of previously published genomic studies in patients with endometriosis, adenomyosis, uterine leiomyoma, endometrial cancer, or RIF supports the hypothesis of the alterations in the aforementioned pathogenetic mechanisms, in all these disorders [25]. Although the genomics studies (transcriptomic approach) did not investigate cellular mechanisms, there is a body of evidence suggesting that the uterine NK cells may play a major role in several of the previously described pathogenetic pathways. Also, there are studies that document the expression of a multitude of genes vital for implantation in endometrial stromal cells co-cultured with supernatants from uterine NK cell cultures [26,27]. The arguments that sustain, at least in part, the association between the NK cell and these pathogenic pathways are given by its regulatory properties expressed by means of chemokines and cytokine production (e.g., IL-8, IL-1β, TNF-α, IFN-g, and VEGF) and its cytotoxic activity, on one hand, but also by its capacity to modulate the expression of endometrial cells or other cells that mediate the immune response (T lymphocytes), on the other hand [26,28]. The involvement of the NK cell in the development of endometriosis is strongly supported by scientific evidence that showed changes in its regulatory function, as well as in its cytotoxic potential [29]. The expression of inhibitory receptor CD94/NKG2A and NK cell expression was increased in patients with endometriosis. The presence of inhibitory phenotypic variants of NK cells in the ectopic endometrial tissue in adenomyosis has been equally reported, leading to the assumption that a lower cytotoxic potential of NK cells may facilitate the onset of adenomyosis/endometriosis [30,31,32]. In addition, in RIF patients the central disfunctions target the regulatory activity, expressed as altered angiogenic components (e.g., decreased VEGF and placental growth factor (PLGF)) [30]. In terms of representativity, the results of a meta-analysis clearly showed that the uterine NK cell number did not differ significantly between patients with or without RIF [31], thus underlining the importance of both the numeric representation and the functionality of NK cells [32].

Most studies focusing on the KIR genotype of uterine NK cells examined its impact on pregnancy and found that it might be associated with pathologies such as preeclampsia, intrauterine growth restriction, and placenta acretta, providing consistent evidence towards the association of the KIR AA variant with pregnancy outcomes [19,33,34]. A few other studies reported results supporting the association between KIR variants and endometriosis [20,29,31]. Some authors reported changes in the number of NK cells in patients with chronic endometritis, but they did not document a correlation between these cells and the onset of the disease [35,36]. Currently, it is accepted that there are no important changes in the number of NK cells at the uterine level, and any diseases with an immune component are rather due to altered cell function.

Our study results support the hypothesis of an association between the KIR AA genotype and chronic endometritis. The overall significantly increased frequency of chronic endometritis in a population group with minimal clinical symptoms was slightly surprising. The link between this disorder and the immune status may be explained by the non-specific factors involved. The association between chronic endometritis and the KIR AA genotype is a relatively recent finding, but it was suggested by previously published research results. Chronic endometritis is an underdiagnosed and probably underestimated condition which is frequently associated with other pathologies, such as polyps and adenomyosis [37,38] and especially endometrial cancer [39] and RIF [40,41,42]. Unlike the acute form, chronic endometritis seems to be less related to the aggressiveness of the germs and more to an alteration of the local flora, combined with a decrease in the immune response. In this respect, a decreased immune response related to the KIR polymorphism may explain a patient’s susceptibility towards this disorder. Chronic endometritis (even asymptomatic forms) was associated with poor reproductive outcomes. Several studies reported lower pregnancy rates in IVF patients with this disorder [31,42,43]. Moreover, it is speculated that the decrease in LIF levels may be one of the underlying pathogenetic pathways [41]. Even if in current practice it is accepted that normal receptivity cannot be characterized by a single factor, changes in LIF levels remain a visible marker of an important impairment at this level [44].

As regards the uterine cytokines, we found higher levels of IL-1β and TNF-α, and lower levels of LIF in patients with chronic endometritis, regardless of the KIR genotype. However, in the KIR AA subgroup, both TNF-α and IL-1β levels were significantly lower as compared to the other variants, suggesting an altered behavior of the NK cell regulatory functions. These subtle changes disclosed in the presence of stressors (e.g., chronic infection) are consistent with the hypothesis of a reduced regulatory capacity of the NK cells, that may be related to the development of chronic endometritis (which has as an inherent clinical consequence, a diminished fertility potential), and other disorders with similar pathogenetic mechanisms (e.g., adenomyosis and endometriosis). The modified levels of TNF-α, IL-1β and LIF cytokines at the uterine level in patients with chronic endometritis could suggest that this disorder may be associated with clinical consequences such as a lower implantation rate, but it may also suggest a potential role of chronic endometritis in the development of other pathologies, including endometrial cancer, as reported also in other studies [37,38,39].

Currently, the recommendation to test for KIR polymorphism all infertile patients would most likely be premature since a relationship of causality between chronic endometritis and uterine pathology has not yet been documented. Finally, the altered production of cytokines associated with uterine disorders in the KIR AA group allows us to expand the area of a potentially altered pathogenetic mechanism from the territory of cellular recognition towards the regulatory mechanism level, opening up a whole new research area.

Our study has several limitations. The study population was enrolled from a single IVF treatment center, and so may not be representative of all Romanian IVF patients. Other major limitations include low sample size and no randomizations, and also the fact that we used multiple hypothesis tests and did not adjust for confounding. Our statistical power to detect modest associations between the KIR AA genotype and endometrial dysfunctions was limited by the small sample size, and we did not adjust for potentially chance findings from statistical testing errors. In addition, we used the cytokine concentration levels in the endometrium washing fluid and the extrapolation of data on the regulatory behavior of the NK cell as proxies to evaluate the impact of the KIR genotype on the endometrium in IVF patients which could have introduced errors into the evaluation, therefore, more comprehensive studies are necessary to confirm this hypothesis.

## 5. Conclusions

Our study results suggest that the KIR AA genotype population subgroups may be more susceptible to developing endometrial disorders such as chronic endometritis. The changes in the behavior of NK cells seem to be subtle and expressed as an altered regulatory pattern. Although the changes are minor, they may trigger obstetric and gynecologic disorders with significant clinical impact. However, larger studies are necessary to confirm the hypothesis that the KIR AA genotype population subgroup may be more susceptible to developing endometrial dysfunctions and, implicitly, other endometrium-derived uterine pathologies. In this regard, the recommendation of an active attitude in the diagnosis process could shorten the time necessary to identify/treat silent disorders such as chronic endometritis, with significant clinical implications (e.g., repeated implantation failure). Also, the implementation in the current clinical practice of specific tests to identify population subgroups more susceptible to developing certain pathologies may contribute greatly to the improvement of therapeutic intervention in terms of safety and efficiency.

## Figures and Tables

**Figure 1 diagnostics-13-03048-f001:**
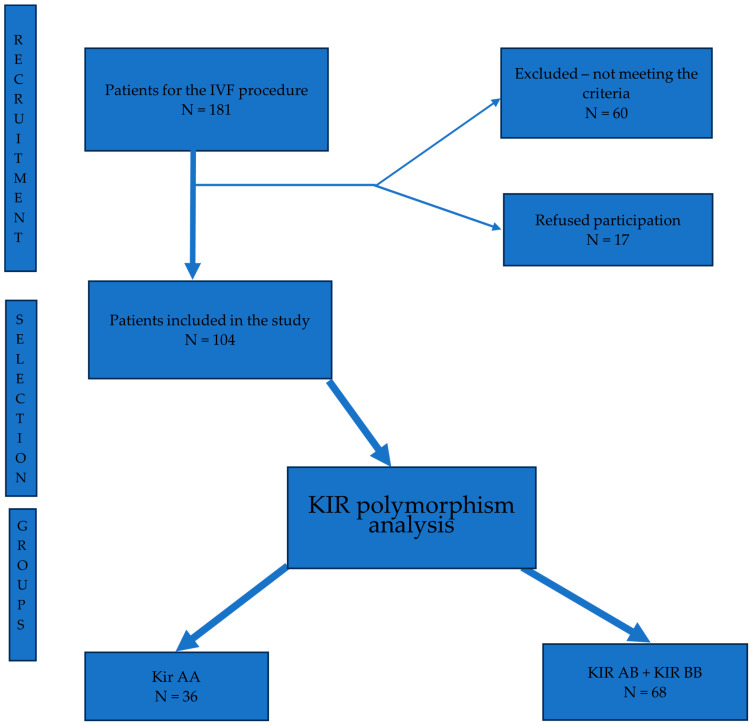
Patient recruitment flow diagram.

**Figure 2 diagnostics-13-03048-f002:**
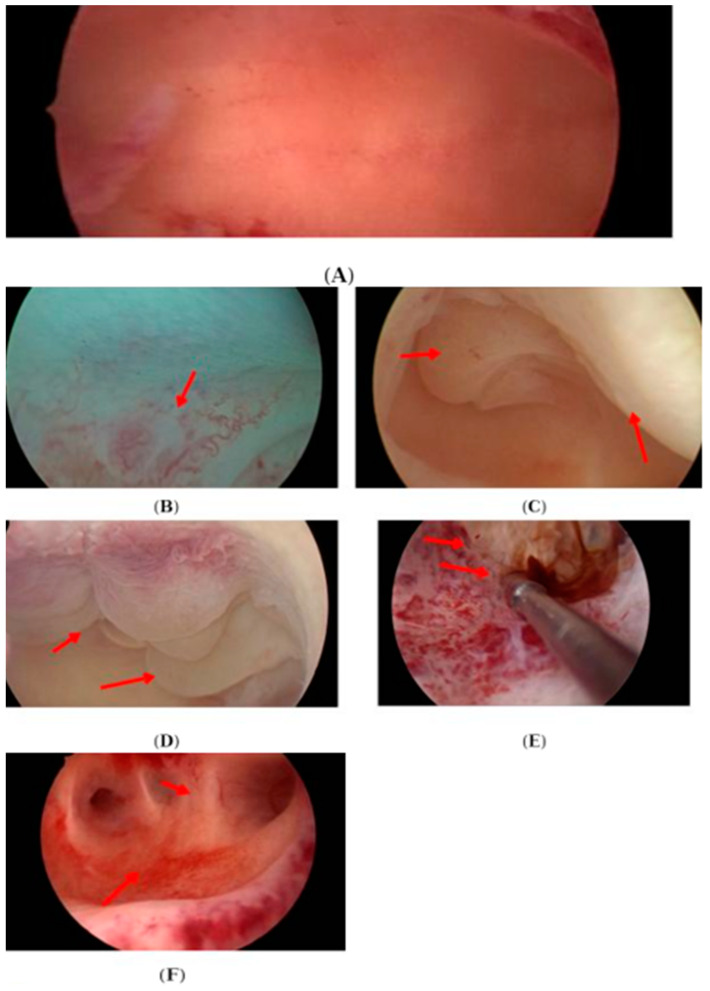
Hysteroscopic pathological findings in study participants. (**A**) Normal aspect. (**B**) Abnormal endometrial vascularisation. (**C**) Elevation of the mucosa and polyps. (**D**) Micropolyps. (**E**) Synechia and adenomyosis. (**F**) Synechia and hypervascularization.

**Table 1 diagnostics-13-03048-t001:** Distribution of demographic, clinical and paraclinical factors among study participants by KIR genotype.

	KIR AA Genotype				KIR BB and AB Genotype				
Factors	n	Mean ± SD (%)	Median	25th–75th %	n	Mean ± SD (%)	Median	25th–75th %	*p*-Value
**Female age (years)**	36	37.3 ± 3.3	38	35.5–40	68	36.7 ± 3.7	37	34–40	0.416
**Female BMI (kg/m^2^)**									0.518
Underweight and Normal (<25)	23	(63.9)			39	(57.3)			
Overweight and Obese (≥25)	13	(36.1)			29	(42.6)			
**Female Smoking Status**									0.486
Current smoker	6	(16.7)			8	(11.8)			
Nonsmoker	30	(83.3)			60	(88.2)			
**Infertility Diagnosis**									
Female factor—tubal	6	(16.7)			11	(16.2)			0.949
Female factor—ovarian	5	(13.9)			12	(17.7)			0.622
Male factor	11	(30.5)			19	(27.9)			0.779
Idiopathic	14	(38.9)			26	(38.2)			0.948
**Infertility duration** **(years)**	36	5.3 ± 2.1	5	3–7	68	4.3 ± 1.8	4	3–5	**0.015**
**No. of previous IVF cycles**	36	2.2 ± 1.1	2	1–3	68	1.5 ± 1.0	1	1–2	**0.002**
**No. of previous abortions**	36	0.4 ± 0.6	0	0–1	68	0.2 ± 0.4	0	0–0	**0.041**
**Clinical symptoms among study participants**									0.933
No	33	(91.7)			62	(91.2)			
Yes	3	(8.3)			6	(8.8)			
**Endometrium thickness (mm)**									0.755
>6 mm	32	(88.9)			59	(86.8)			
<6 mm	4	(11.1)			9	(13.2)			
**No. of abnormal hysteroscopic findings**	36	2.0 ± 1.5	2	1–3	68	1.0 ± 0.9	1	0–2	**0.0005**
**Chronic endometritis** **(CD 138)**									**0.002**
**No**	12	(33.3)			44	(64.7)			
**Yes**	24	(66.7)			24	(35.3)			
**TNF-α (pg/mL)**	36	65.6 ± 37.2	72	28–93.5	68	65.5 ± 54.3	37	22–106.5	0.557
**IL-1β (pg/mL)**	36	80.9 ± 33.4	85	54–105.5	68	89.8 ± 61.4	58.5	40–151.5	0.848
**LIF (pg/mL)**	36	208.5 ± 45.8	203	175.5–241	68	222.5 ± 40.3	221	199.5–251.5	0.122

Abbreviations: BMI, body mass index; LIF, Leukemia Inhibitory Factor; IL-1β, Interleukin 1 beta; SD, Standard deviation; TNF-α, Tumor Necrosis Factor α; 25th, 25 percentile; 75th, 75 percentile; in bold, *p* < 0.05.

**Table 2 diagnostics-13-03048-t002:** Distribution of uterine cytokine median concentration levels among study participants, by KIR genotype and chronic endometritis diagnosis, and comparative analysis of cytokine median concentration levels between KIR genotype groups with and without chronic endometritis diagnosis.

Genotype	KIR AA				KIR AB and BB				*p*-Value Median Test	
Chronic Endometritis	No		Yes		No		Yes		No	Yes
	Median	25th–75th %	Median	25th–75th %	Median	25th–75th %	Median	25th–75th %		
TNF-α (pg/mL)	23.5	14.5–30	82.5	61.5–104	26.5	17.5–35.5	130.5	104–152.5	0.852	**0.001**
IL-1β (pg/mL)	42	32.5–63	100.5	84.5–116	45	35–58	162	146.5–186.5	0.852	**0.000**
LIF (pg/mL)	241	216.5–268	197.5	156.5–213	230.5	203–258.5	206	177.5–236.5	0.103	0.773

Abbreviations: KIR, Killer Cell Immunoglobulin-like Receptors; LIF, Leukemia Inhibitory Factor; IL-1β, Interleukin 1 beta; TNF-α, Tumor Necrosis Factor α; 25th, 25 percentile; 75th, 75 percentile; in bold, *p* < 0.05.

**Table 3 diagnostics-13-03048-t003:** Correlations between clinical and paraclinical factors and chronic endometritis among study participants.

Clinical and Paraclinical Factors	Correlation Coefficient and *p*-Value					
	Chronic Endometritis	TNF-α	IL-1β	LIF	Hysteroscopic Findings	Previous Abortions
TNF-α	*0.783* **0.000**					
IL-1β	*0.798* **0.000**	*0.782* **0.000**				
LIF	*−0.427* **0.000**	*−0.380* **0.0001**	*−0.277* **0.004**			
Hysteroscopic findings	*0.524* **0.000**	*0.278* **0.004**	*0.250* **0.010**	*−0.188* 0.055		
Previous abortions	*0.137* 0.165	*0.065* 0.515	*−0.003* 0.977	*−0.009* 0.923	*0.231* **0.018**	

Abbreviations: LIF, Leukemia Inhibitory Factor; IL-1β, Interleukin 1 beta; TNF-α, Tumor Necrosis Factor α; in italics, correlation coefficients; in bold, *p* < 0.05.

## Data Availability

Not applicable.

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
