# Peer review of "Killer Cell Immunoglobulin-like Receptor Genotypes and Reproductive Outcomes in a Group of Infertile Women: A Romanian Study"

_diagnostics, 2023, doi:10.3390/diagnostics13193048_

Round 1

Reviewer 1 Report

Figures are not correctly arranged, especially the flow diagram. In Figure 1, better add normal phenotype for readers' convenience. Also, Figure 1 is not set perfectly, and some images are overlapping. For readers' convenience, better add some representative images of ultrasound. Please provide detailed figure legends. Why KIR only? Are there any reasons for excluding other possible genes/genotypes? What is the novelty of your study? Do not need to describe the results in the discussion section; discuss them logically by comparing them with previous studies. You can highlight critical findings/points in the discussion for comparison or reference. 

I've noticed some grammatical and other language errors. 

Author Response

We thank the Reviewer for the suggestions and constructive comments. The critiques provided have helped us to clarify and improve our work. Please find below our responses, in Italics, labeled and bolded as “Response”, for all the questions raised by the Reviewer. Revisions made in response to the reviewer’s comments and suggestions are marked as track changes in our resubmitted manuscript.

Responses to the Comments from Reviewer #1

Comment 1. Figures are not correctly arranged, especially the flow diagram. In Figure 1, better add normal phenotype for readers' convenience. Also, Figure 1 is not set perfectly, and some images are overlapping. For readers' convenience, better add some representative images of ultrasound. Please provide detailed figure legends.

Response: Thank you for the suggestions and constructive comments. As the Reviewer suggested we revised the flow diagram and Figure 1 accordingly, arranged them correctly and included the Reviewer’s suggestions (normal phenotype,representative images of ultrasound, detailed figure legends) in our resubmitted manuscript (pages 6 and 12). Also, we renamed the flow diagram as Figure 1 as requested by the Reviewer #2.

Comment 2. Why KIR only? Are there any reasons for excluding other possible genes/genotypes? What is the novelty of your study?

Response: Thank you for the constructive comments. Killer cell immunoglobulin-like receptors (KIR) are a specific group of NK receptors that recognizes the major histocompatibility complex (MHC) class. Taking into consideration that one of the main functions of NK cell is to discriminate between the healthy autologous cells and the abnormal cells, this type of receptors (KIR) can be considered essential, particularly for the NK cells. In addition, as compared to many other polymorphisms that are just a subject for research, the KIR polymorphism has been consistently associated to clinical conditions also in other pathology (organ transplant). Moreover, despite an impressive number of genes involved, two basic groups of haplotypes are stand out, based on their activity (an inhibitory or activating signal): haplotypes A and B. We choose to analyze the KIR system in this study because of its importance in NK function but also due to its clinical impact. As regards other NK genes, most of them have not yet been widely tested and they still require a detailed investigation of the potential clinical impact. The novelty of this research is the fact that the type A KIR polymorphism may have an impact on fertility, either by the means of an abnormal cytokine production or by an abnormal local immune tolerance which ultimately associates chronic endometritis.

Comment 3. Do not need to describe the results in the discussion section; discuss them logically by comparing them with previous studies. You can highlight critical findings/points in the discussion for comparison or reference. 

Response: Thank you for the suggestions. As the Reviewer suggested, in our resubmitted manuscript, we revised the Discussion section accordingly (pages 13-16).

Comment 4. I've noticed some grammatical and other language errors. 

Response: Thank you. As the Reviewer suggested, before resubmission, we carefully revised our manuscript to correct English language errors.

Reviewer 2 Report

The manuscript cannot be accepted as it needs extensive rewriting, details can be found in the attached document.  

Although the language in general is acceptable, there are some syntax errors that may mislead or confuse the readers if not corrected.  

Author Response

Please see the attachment with our responses provided point by point.

Round 2
